

# Characteristics and outcomes of hemodialysis patients with COVID-19: a retrospective single center study

Yongwen Luo[1,*], Junli Li[2,*], Zhifen Liu[3], Heping Yu[4], Xiang Peng[5] and Cheng'an Cao[5]

[1] Department of Urology, Zhongnan Hospital of Wuhan University, Wuhan, China
[2] Institute of Laboratory Animal Sciences, Peking Union Medical College, Beijing, China
[3] Department of Nephrology, Wuhan Fourth Hospital, Wuhan, China
[4] Department of Thyroid and Breast Surgery, Wuhan Fourth Hospital, Wuhan, China
[5] Department of Neurology, Wuhan Fourth Hospital, Wuhan, China
* These authors contributed equally to this work.

## ABSTRACT

**Background:** The coronavirus 19 (COVID-19) pandemic has heightened the threat to the health and lives of patients with comorbid diseases. Infection by COVID-19 is especially detrimental to patients on hemodialysis. In this study, we evaluated the clinical characteristics, laboratory findings, treatments and prognoses of hemodialysis patients with COVID-19.

**Methods:** A total of 16 hemodialysis patients with COVID-19 were recruited from Wuhan Fourth Hospital from 5 February to 20 March 2020 for a retrospective, single-center study. A total of 62 non-dialysis patients with COVID-19 were the control group. We collected data on the clinical characteristics, laboratory findings, treatments, and clinical outcomes of patients affected by the virus.

**Results:** Hemodialysis patients with COVID-19 had a lower incidence of fever ($P = 0.001$) and relatively higher incidence of pre-admission comorbidities and shortness of breath than non-dialysis patients with COVID-19 (75% vs. 61%, $P = 0.467$ 50% vs. 33.87%, $P = 0.248$ ). Hemodialysis patients had lower levels of hemoglobin ($P < 0.001$), white blood cell counts ($P = 0.015$), neutrophils ($P = 0.016$), AST ($P = 0.037$), ALT ($P < 0.001$) and procalcitonin ($P < 0.001$), and higher levels of D-dimer ($P < 0.001$) and thrombin time ($P < 0.001$). Hemodialysis patients had a higher incidence of pulmonary effusion, cord-like high-density shadows, pleural thickening, and atelectasis ($P < 0.05$). Hemodialysis patients also had relatively higher rates of mortality and prolonged hospital stays compared with the control group.

**Conclusions:** Hemodialysis patients typically present with multiple comorbidities and are considered to be a high-risk group for COVID-19 infections. Hemodialysis patients with COVID-19 may have prolonged hospital stays and unfavorable prognoses and should be closely monitored.

Corresponding author
Cheng'an Cao, cca24@126.com

## INTRODUCTION

The coronavirus 2019 (COVID-19) pandemic caused by severe acute respiratory syndrome coronavirus 2 (SARS-CoV-2) has created a major public health issue (*CDC COVID-19 Response Team, 2020*). There have been 24,257,989 confirmed cases and 827,246 deaths reported across 209 countries or regions since 28 August 2020 (*WHO, 2020b*). Previous studies have shown that older individuals, particularly those with chronic comorbidities, are generally more susceptible to infections by COVID-19 (*Chen et al., 2020*; *Wu & McGoogan, 2020*). A total of 41.3% of COVID-19 cases occurred due to hospital-related transmissions (*Wang et al., 2020a*). Hemodialysis patients require hospital stays for treatments multiple times per week and are already immunocompromised due to uremia. Many of these patients are also elderly and have comorbidities including coronary disease, hypertension, diabetes and lung disease that are associated with unfavorable outcomes in patients with COVID-19 (*Zhou et al., 2020*). Hemodialysis patients are therefore more highly susceptible to infection by COVID-19 than the general population (*Wang et al., 2020b*; *Xiong et al., 2020*). There is a particular interest in the clinical features and outcomes of hemodialysis patients infected with COVID-19. Early research has found that the clinical symptoms of COVID-19, including cough and fever, were less common in patients on hemodialysis but that hemodialysis patients with COVID-19 had a higher risk of death than the general population (*Tortonese et al., 2020*; *Wu et al., 2020*; *Yang et al., 2020*). Therefore, we evaluated the clinical characteristics, laboratory findings, radiological characteristics, treatments, and outcomes of 16 hemodialysis patients with confirmed COVID-19 infections in order to provide an insight into the clinical assessment and management of hemodialysis patients with COVID-19.

## PATIENTS AND METHODS

### Study design and participants

We recruited hemodialysis patients with confirmed COVID-19 infections from Wuhan Fourth Hospital, Tongji Medical College, Huazhong University of Science and Technology, Wuhan, China from 5 February to 20 March 2020 for our retrospective, single-center study. Wuhan Fourth Hospital was designated as the treatment center for hemodialysis patients with COVID-19 during the outbreak and patients with confirmed or suspected COVID-19 infections were centrally admitted. All hemodialysis patients enrolled in this study had confirmed COVID-19 infections diagnosed according to the interim guidance by the WHO (*CDC, 2020*; *WHO, 2020a*). We compared the results from hemodialysis patients with a control group of non-dialysis COVID-19 patients at the same hospital. We adjusted for age and sex and matched non-hemodialysis COVID-19 patients for sex and age by randomly selecting each hemodialysis patient with COVID-19 according to previously reported methods (*Cortes Garcia et al., 2012*; *Lovshin et al., 2018*). Our study was approved by the Institutional Review Board (IRB) at Wuhan Fourth Hospital (IRB approval number: KY 2020-032-01). Informed consent was waived as part of the public health outbreak investigation.

## Data collection

Two doctors in our team gathered clinical information from electronic medical records. Patient information included epidemiology, demographics, medical history, laboratory findings, comorbidities, treatment regimens (antiviral, antibiotic, corticosteroid therapies, immune glucocorticoid therapy and respiratory support), length of hospital stay, and clinical outcomes. Data that was missing from the medical records were obtained through direct communication with attending doctors or the patients themselves.

## Definitions

Discharge criteria for patients were defined as: a normal body temperature for more than 3 days, significant improvement in respiratory symptoms, improved chest CT imaging indicating reduced inflammation, and two consecutive negative nucleic acid tests throat swabs with at least a 1-day interval between tests. A comprehensive evaluation was made by an expert team to determine whether the patient could be discharged.

## Statistical analysis

Categorical variables were expressed as numerical values (%) and continuous variables were presented as median with interquartile ranges (IQR) in the descriptive statistics. We compared the means for discrete variables using independent Student's $t$ tests when the data were normally distributed. The Mann-Whitney test was used for data not normally distributed. Proportions for categorical variables were compared using the $\chi^2$ test, although the Fisher's exact test was used when the sample sizes were small. All statistical analyses were performed using SPSS software (version 21.0) and $P < 0.05$ was considered statistically significant.

# RESULTS

## Clinical characteristics

A total of 16 hemodialysis patients with COVID-19 were included in our study. The clinical characteristics of the 16 patients are shown in Table 1. The median patient age was 61 years (interquartile range 54–78 years), and 8 of the 16 patients were female. None of the patients had exposure to the Huanan seafood market that appeared to be the epicenter of the COVID-19 outbreak. All hemodialysis patients studied had a history of being in contact with the epidemic area and of being in contact with patients who had a fever. Twelve cases (75%) had comorbidities including coronary disease, hypertension and diabetes. The most common initial symptoms at admission were: cough (75%), shortness of breath (50%), fatigue (50%) and fever (43.75%). The median incubation period was 9 days (interquartile range 4.3–12 days). Hemodialysis patients presented with a lower incidence of fever compared with non-dialysis patients ($P = 0.001$). Hemodialysis patients also had higher incidence of preadmission comorbidities and shortness of breath, although the difference was not significant.

**Table 1 Clinical characteristics of hemodialysis patients with COVID-19.**

| Characteristics | No. (%) | | P value |
|---|---|---|---|
| | Hemodialysis patients | Non-dialysis patients | |
| Age (years), median (IQR) | 61 (54–78) | 62 (50–70) | 0.439 |
| Male | 8 (50) | 33 (53.2) | 0.755 |
| Contact history of epidemic area | 16 (100) | 62 (100) | >0.99 |
| Preadmission comorbidities | 12 (75) | 38 (61) | 0.467 |
| Hypertension | 11 (68.75) | 19 (30.65) | **0.012** |
| Diabetes | 3 (18.75) | 11 (17.74) | >0.99 |
| Cardiovascular diseases | 4 (25) | 4 (6.45) | 0.086 |
| Cerebrovascular diseases | 0 (0) | 2 (3.23) | >0.99 |
| Malignancy | 1 (6.25) | 2 (3.23) | >0.99 |
| COPD | 0 (0) | 2 (6.45) | >0.99 |
| Tuberculosis | 0 (0) | 2 (3.23) | >0.99 |
| Pneumonia related manifestations | | | |
| Cough | 12 (75) | 45 (72.58) | 0.699 |
| shortness of breath | 8 (50) | 19 (33.87) | 0.248 |
| Fatigue | 8 (50) | 42 (67.74) | 0.305 |
| Fever | 7 (43.75) | 54 (87.10) | **0.001** |
| Diarrhoea | 2 (12.5) | 11 (17.74) | 0.900 |
| Myalgia | 2 (12.5) | 23 (37.10) | 0.275 |
| Headache | 1 (6.25) | 2 (3.22) | >0.99 |
| Sore throat | 1 (6.25) | 3 (4.84) | >0.99 |
| Incubation period (days), median (IQR) | 9 (4.3–12) | 8 (5–12) | 0.674 |

**Note:**
Data are reported as $n$ (%) or median (IQR). The $P$ value represents the difference between hemodialysis and non-dialysis patients. $P$ value < 0.05 was considered significant difference (bold font).

## Radiological and laboratory findings

A total of 16 patients (100%) presented with abnormal chest scans at admission. The scans revealed typical signs of infection for COVID-19 in hemodialysis patients (Fig. 1), such as bilateral patchy shadowing (100%) and ground-glass opacities (62.5%) (Table S1). Hemodialysis patients presented with a higher incidence of pulmonary effusion ($P < 0.001$), cord-like high-density shadows ($P < 0.001$), pleural thickening ($P = 0.0046$), atelectasis ($P = 0.0046$) and consolidation of lung tissues ($P = 0.053$) when compared with non-dialysis patients (Table 2). A total of 15 (93.75%) hemodialysis patients had anemia at admission and 14 (92.86%) had coagulopathy with elevated D-dimer, while all hemodialysis patients presented with hypoproteinemia (100%) and lymphopenia (100%). Infection-related biomarkers, including procalcitonin and C-reactive protein, were also abnormal in almost all patients (Table S1). Hemodialysis patients had lower levels of hemoglobin ($P < 0.001$), white blood cell counts ($P = 0.015$), neutrophils ($P = 0.016$), AST ($P = 0.037$), ALT ($P < 0.001$) and procalcitonin ($P < 0.001$), and higher levels of D-dimer ($P < 0.001$) and thrombin time ($P < 0.001$) when compared with non-dialysis patients (Table 2).

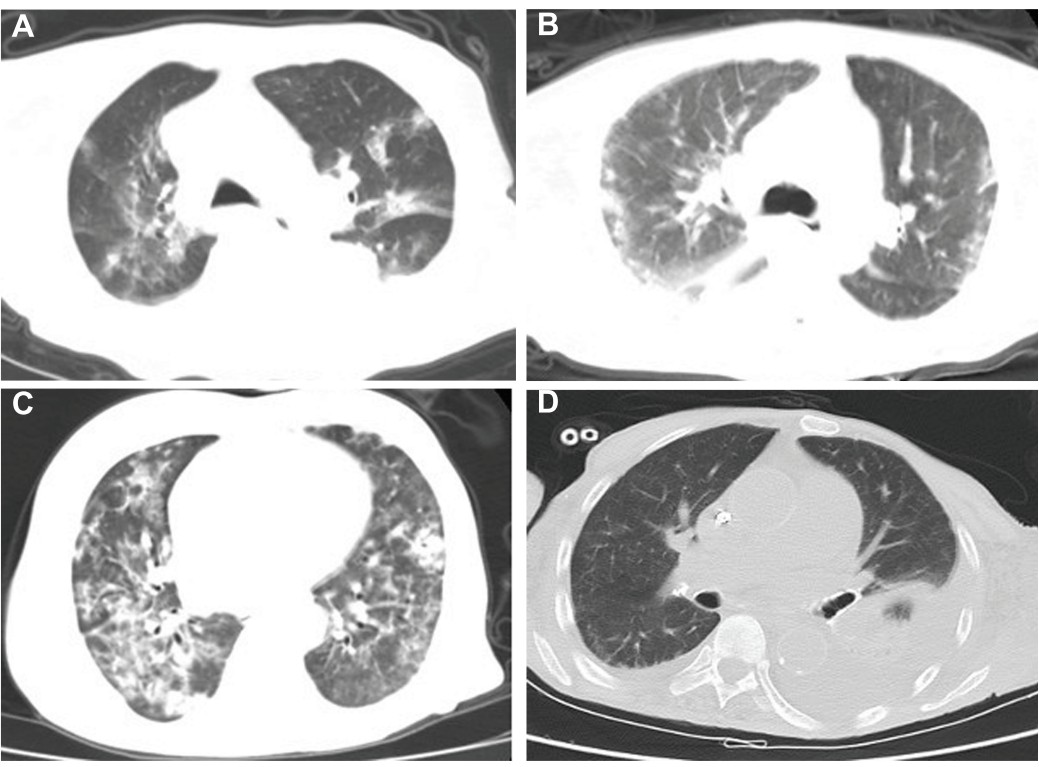

**Figure 1 Chest CT imaging of four hemodialysis patients.** (A–D) Typical signs of infection for COVID-19 of four hemodialysis patients.          

## Treatment and outcomes

All patients received treatment according to the Chinese Diagnosis and Treatment Protocol for COVID-19 (sixth version) (*CDC, 2020*). 16 patients received oxygen therapy in while in isolation and 14 patients received antiviral treatment, including Arbidol (0.2 g three times daily, orally), Lopinavir and Ritonavir tablets (0.5 mg twice daily, orally), Oseltamivir (75 mg twice daily, orally) or Ribavirin (500 mg once per day, intravenously). Traditional Chinese medicines, such as Lianhuaqingwen capsules and Toujiequwen Keli, were also given. A total of 14 patients (87.5%) were treated with antibiotics, including Amoxicillin clavulanate potassium, Levofloxacin, Moxifloxacin, Cephalosporin, Ceftriaxone, Cefoxitin, or Cefoperazone sulbactam. Nine patients (56.25%) received a single antibiotic treatment and five patients (31.25%) received a combination of treatments. Two patients (12.5%) also received corticosteroid treatment and one patient (6.25%) received intravenous immunoglobulin therapy (Table S2). There was no difference between hemodialysis and non-dialysis patients in terms of treatment modality (Table S3). Hemodialysis was performed with the usual frequency in an isolation room designated for COVID-19 patients. We analyzed the clinical outcomes of the study participants at the end of the follow-up period on 5 April 2020. The median period of hospitalization was 21 days (interquartile range, 15.5–30.25 days). A total of 13 patients recovered and were discharged from the hospital and three patients died, resulting in a mortality rate of 18.75%. Hemodialysis patients showed higher mortality and a prolonged

**Table 2 Laboratory results of Hemodialysis patients with COVID-19.**

| Characteristics | No. (%) | | P value |
|---|---|---|---|
| | Hemodialysis patients | Non-dialysis patients | |
| Admission radiologic findings (Chest CT) | | | |
| Bilateral patchy shadowing | 16 (100) | 58 (93.5) | 0.683 |
| Ground-glass opacities | 10 (62.5) | 47 (75.8) | 0.451 |
| Pulmonary effusion | 8 (50) | 2 (3.23) | **<0.001** |
| Cord high density shadows | 7 (43.75) | 4 (6.45) | **<0.001** |
| Pleural thickening | 4 (25) | 1 (1.6) | **0.0046** |
| Atelectasis | 4 (25) | 1 (1.6) | **0.0046** |
| Consolidation of lung | 2 (12.5) | 0 (0) | 0.053 |
| Blood routine | | | |
| Hemoglobin (g/L) | 91.5 (79.25–102.5) | 127.5 (115–141.75) | **<0.001** |
| White blood cell count (×10^9/L) | 4.29 (3.56–6.69) | 6.22 (4.66–8.38) | **0.015** |
| Neutrophil (×10^9/L) | 3.19 (2.47–5.3) | 5.18 (3.27–7.44) | **0.016** |
| Lymphocyte count (×10^9/L) | 0.65 (0.51–0.91) | 0.74 (0.40–1.17) | 0.656 |
| Coagulation function | | | |
| APTT | 31.5 (26.6–43.3) | 35.5 (32.13–38.32) | 0.290 |
| Prothrombin time (s) | 12.2 (11.05–14.13) | 12.9 (12.3–13.33) | 0.133 |
| TT | 19.5 (18.9–20.98) | 15.7 (14.7–16.5) | **<0.001** |
| D-dimer (mg/L) | 1.84 (0.88–4.55) | 0.49 (0.24–1.27) | **<0.001** |
| Blood biochemistry | | | |
| AST (U/L) | 20 (13–24) | 25.5 (17–42.25) | **0.037** |
| ALT (U/L) | 10 (6.25–13.5) | 28 (16–42.25) | **<0.001** |
| Serum creatinine (μmol/L) | 1067.5 (855.08–1392.23) | 70.5 (55.75–82.25) | **<0.001** |
| Infection-related biomarkers | | | |
| C-reactive protein (mg/L) | 39.3 (23.85–87.53) | 44.74 (20.73–74.20) | 0.782 |
| Procalcitonin (ng/mL) | 0.85 (0.44–2.79) | 0.04 (0.04–0.06) | **<0.001** |

Note:
Data are reported as $n$ (%) or median (IQR). The $P$ value represents the difference between hemodialysis and non-dialysis patients. $P$ value < 0.05 was considered significant difference (bold font).

hospital stay, although the results were not significantly different ($P = 0.427$ and $P = 0.077$, respectively).

## DISCUSSION

The effect of COVID-19 on hemodialysis patients is of great interest to clinicians. Our retrospective, single-center study details the clinical characteristics and outcomes of hemodialysis patients infected with SARS-CoV-2.

We found that hemodialysis patients presented with similar clinical symptoms as patients with COVID-19 from the general population. Symptoms included: fever, cough, sore throat, shortness of breath, myalgia, headache, fatigue, and diarrhea (*Wu & McGoogan, 2020*; *Xiong et al., 2020*). The incubation period lasted approximately 1–14 days in both groups. A previous study reported that 18.7% of non-dialysis patients had shortness of breath (*Xiong et al., 2020*), but 50% of hemodialysis patients experienced

shortness of breath, which may be attributed to several factors. Firstly, half of the hemodialysis patients (50%) had pulmonary effusion caused by hypoproteinemia due to chronic kidney disease and inadequate hemodialysis. Secondly, some hemodialysis patients experienced extended periods of lung inflammation and pleural thickening due to long-term accumulation of fluid in the lungs. Almost all hemodialysis patients had anemia, causing an increased burden on the heart and shortness of breath. Hemodialysis patients also had a lower incidence of fever, which may be related to immunosuppression caused by uremia.

All hemodialysis patients presented on chest CT with typical signs of a viral infection, including ground-glass opacities and bilateral patchy shadowing. Many patients on maintenance hemodialysis had pulmonary effusion due to inadequate hemodialysis and chronic lung inflammation caused by uremia and pulmonary effusion, which may contribute to CT features such as cord-like high-density shadows, pleural thickening, atelectasis, and pulmonary fibrosis. Healthcare-associated pneumonia (HDAP) is commonly contracted by hemodialysis patients (*Carratalà & Garcia-Vidal, 2008*; *Lee & Moon, 2016*). Therefore, these CT features may make the diagnosis of COVID-19 infection more difficult in hemodialysis patients, compared with non-dialysis patients.

The majority of the hemodialysis patients with COVID-19 in our study were middle-aged and senile; 12 patients (75%) had comorbidities, including coronary disease, hypertension, and diabetes. Most hemodialysis patients were immunocompromised due to uremia (*Betjes, 2013*; *Kim et al., 2017*; *Syed-Ahmed & Narayanan, 2019*), which may have led to longer hospital stays and higher mortality than for those in the general population.

Almost all hemodialysis patients had anemia, hypoproteinemia, and lymphopenia coagulopathy with elevated D-dimer at admission, increasing the risk of an unfavorable prognosis. This is keeping with the finding of a previous study that reported that severe lymphopenia and elevated D-dimer were risk factors for the prognosis of patients with COVID-19 (*Zhou et al., 2020*). However, in hemodialysis patients, elevated D-dimer may be partly caused by uremia and hemodialysis, therefore we could not simply consider D-dimer levels as risk factors for mortality in hemodialysis patients with COVID-19, as in the general population.

SARS-CoV-2, which is responsible for COVID-19, belongs to the β-type coronavirus, is enveloped, has round or oval particles, and is often polymorphic, with a diameter of 60–140 nm (*Shang et al., 2020*). Current research has shown that SARS-CoV-2 has more than 79% homology with the severe acute respiratory syndrome (SARS) coronavirus, and the S proteins of SARS-CoV-2 and SARS-CoV, have a sequence similarity of approximately 77% (*Lu et al., 2020*; *Yuan et al., 2020*). However, clinical studies have revealed many differences between SARS-CoV-2 and SARS coronavirus. For example, SARS-CoV-2 has higher infectivity and lower lethality compared with SARS coronavirus (*Guan et al., 2020*; *Park et al., 2020*; *Xu et al., 2020*; *Zhang et al., 2020*). A study on severe acute respiratory syndrome reported that dialysis patients had similar clinical features and mortality rates as non-dialysis patients, but dialysis patients tended to present with less pronounced symptoms and had a much longer hospital stay compared with

non-uremic patients (*Kwan et al., 2004*). The dialysis patients we studied tended to have a lower incidence of fever but a higher incidence of shortness of breath and unfavorable clinical prognosis, which may be due to the greater differences in the pathogenic mechanisms of the two viruses. Additional research is needed to validate these findings.

Our study was limited by its small sample size and single center demographics. A systematic and comprehensive study should be conducted to include a larger sample size across multiple centers to better assess the effects of COVID-19 infections in hemodialysis patients. In addition, we were only able to collect the information from hemodialysis patients with COVID-19, but we were not able to compare the clinical characteristics of hemodialysis patients with or without SARS-CoV-2, nor can we describe the epidemiology of COVID-19 infections in hemodialysis patients. However, our study details fundamental information about the characteristics of COVID-19 in this patient population.

In summary, we described the clinical characteristics and outcomes of hemodialysis patients with COVID-19 infections. Hemodialysis patients are a high-risk group for infection by COVID-19. They presented with similar clinical characteristics to those who were not on hemodialysis. Hemodialysis patients had multiple comorbidities and worse physical conditioning, prolonged hospital stays, and unfavorable clinical prognoses. Therefore, hemodialysis patients should be monitored intensively for COVID-19 infections.

## Author contributions

Dr. Luo and Cao had full access to all the data in the study and takes responsibility for the integrity of the data and the accuracy of the data analysis. Study concept and design: J.L. Li, C.A. Cao, Y.W. Luo. Acquisition, analysis, or interpretation of data: Y.W. Luo, Z.F. Liu, J.L. Li, X. Peng. Drafting of the manuscript: J.L. Li, Y.W. Luo. Critical revision of the manuscript for important intellectual content: C.A. Cao, X. Peng. Statistical analysis: J.L. Li, Y.W. Luo. Administrative, technical, or material support: J.L. Li, Z.F. Liu, X. Peng. Study supervision: C.A. Cao.

## ACKNOWLEDGEMENTS

We thank the patients and their family members for their participation in our study.

### Funding

The authors received no funding for this work.

### Competing Interests

The authors declare that they have no competing interests.

### Author Contributions

- Yongwen Luo conceived and designed the experiments, authored or reviewed drafts of the paper, and approved the final draft.
- Junli Li analyzed the data, prepared figures and/or tables, authored or reviewed drafts of the paper, and approved the final draft.
- Zhifen Liu performed the experiments, analyzed the data, authored or reviewed drafts of the paper, and approved the final draft.
- Heping Yu conceived and designed the experiments, performed the experiments, prepared figures and/or tables, and approved the final draft.
- Xiang Peng analyzed the data, prepared figures and/or tables, and approved the final draft.
- Cheng'an Cao conceived and designed the experiments, performed the experiments, prepared figures and/or tables, authored or reviewed drafts of the paper, and approved the final draft.

## Human Ethics

The following information was supplied relating to ethical approvals (i.e., approving body and any reference numbers):

This study was approved by institutional review board (IRB) at Wuhan Fourth Hospital (IRB approval number: KY 2020-032-01).

## Data Availability

Raw data is available in the Supplemental Files.

## Supplemental Information

Supplemental information for this article can be found online at http://dx.doi.org/10.7717/peerj.10459#supplemental-information.

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
