# Peer review of "Characteristics and outcomes of hemodialysis patients with COVID-19: a retrospective single center study"

_PeerJ, doi:10.7717/peerj.10459_

## Round 0.1 · original submission · Major Revisions

The paper has major issues that need to be addressed seriously. Although I agree with the clinical relevance of this study, without substantial revisions, it is difficult to publish it.

·

Basic reporting

The article is written in clear English language with an unambiguous text, but there are grammatical errors as well as noticeable mistakes in sentence structuring and article placements (a/an/the).

The article includes a sufficient introduction and background with proper referencing.

The structure is conformed to an acceptable format of standard sections.

Figures and tables are labelled correctly and relevant to the content of the article.

More detailed information is needed to describe the statistical analysis. For instance what statistical test was used to calculate the p-value , what is the cutpoint p-value for statistical significance.

Appropriate raw data have been made available in accordance with the Data Sharing policy.

The article is self-contained with relevant results to hypotheses, but there are clear limitations in the study design.

Experimental design

The article is original primary research within the aims and scope of this journal. The research question is well-defined, relevant, and meaningful.

Methods need further explanation, especially statistical analysis, and the rationale behind the clear difference in number between hemodialysis patients and non-dialysis controls, which I think had an impact on the calculated p-values.

Validity of the findings

Conclusions are well-stated and linked to the original research question. Limitations of the study are acknowledged. There is a need for multi-center studies with a larger number of patients to confirm the results reached and confirm their validity.

Underlying data have been provided, but I need more information about the statistical tests used to calculate p-values and determine statistical significance.

Additional comments

I think that your article need more editing in term of grammar and sentence structuring. I will advise you to include more details on the statistical tests used and how statistical significance was defined.

A few suggested corrections:

In line 39 - COVID-19 is now considered a global pandemic by the World Health Organization.
In line 44 - As hemodialysis patients need admission to hospital for dialysis.
In lines 52 to 55 - We aim to provide an insight on the clinical assessment and management of hemodialysis patients with COVID-19 infection through describing their clinical characteristics, laboratory
findings, radiological characteristics, treatment, and outcomes.

In line 62 - is a well-established tertiary hospital that has been operational for 150 years.

In line 97/98 - All patients had a contact history with febrile patients within the epidemic area.

In line 109 - hemodialysis patients had a higher incidence of pulmonary effusion.

In line 148 - Hemodialysis patients had clinical symptoms similar to other confirmed COVID-19 cases.

·

Basic reporting

1. I think that English and clarity, in general, should be revised. Indeed, mainly the first part of the abstract (there are two different abstracts uploaded!), the introduction and the methods section are unclear in some points.
For example, looking at the “definitions” paragraph, it is not clear if the patients were discharged while still infected or not.

Experimental design

2. There are data on time to negative SARS-COV2 throat swabs? There were differences between the group? In my experience, I noticed that HD patients often present a long-lasting infection
3. I think that many of the differences observed between COVID+ HD patients vs non-HD were not due to COVID, but to HD condition per sè. The authors recognize this issue in the discussion, but I think that it should be further reinforced (for example this consideration applies also to anaemia).
4. Can we have some data on HD treatment performed (HD, HDF??)
5. How was the incubation period defined?

Validity of the findings

6. Finally, and most importantly, I think that the data here presented, maybe because of the small sample of patients, do not fully support (at least statistically) the authors' conclusions about an increased mortality rate and hospital stay in HD patients. So, I suggest, if it is not possible to increase the number of patients, to be more prudent with the conclusions.

Additional comments

Dear Authors, as you can see from my comments, I have some remarks and suggestions to improve the paper before considering for publication.
In particular, I think that the clarity of the language, the definitions of clinical criteria of infection and the interpretations of the results should be revised to make the paper more readable and suitable.

Reviewer 3 ·

Basic reporting

The logic of this manuscript is not clear enough. In additional, it is a big problem in the written English of this manuscript, even the author and department info. were wrongly written, such as "Nail and breast surgery Department".

Experimental design

1. Hemodialysis patients were a high-risk group of SARS-coV-2 infections. As shown in PubMed, more than 70 papers concerning COVID-19 in Hemodialysis patients have been published since the outbreak of COVID-19. However, the authors claimed several times that "Currently, the research about hemodialysis patients with COVID-19 is scarce. 143 This study is one of the first to report clinical characteristics and outcomes exhaustively". The authors did not search in Pubmed? Compared with these published papers, the current manuscript added little to this area.
2. It is a single-center study with only 16 COVID-19 hemodialysis patients enrolled, which is not representative enough. The data provided is not convincing due to the small sample size.
3. The diagnostic criteria was from WHO, but the treatment was using Chinese guideline 6th edition?
4. Table 3 makes no sense because of the small size of the patients enrolled.

Validity of the findings

No comment.

Additional comments

1. Hemodialysis patients were a high-risk group of SARS-coV-2 infections. As shown in PubMed, more than 70 papers concerning COVID-19 in Hemodialysis patients have been published since the outbreak of COVID-19. However, the authors claimed several times that "Currently, the research about hemodialysis patients with COVID-19 is scarce. 143 This study is one of the first to report clinical characteristics and outcomes exhaustively". The authors did not search in Pubmed? Compared with these published papers, the current manuscript added little to this area.
2. It is a single-center study with only 16 COVID-19 hemodialysis patients enrolled, which is not representative enough. The data provided is not convincing due to the small sample size.
3. The diagnostic criteria was from WHO, but the treatment was using Chinese guideline 6th edition?
4. Table 3 makes no sense because of the small size of the patients enrolled.

---

## Round 0.2 · Major Revisions

The paper still has a lot of issues. It seems that the authors need to re-edit the paper carefully. In addition, there are some basic problems in the writing, organization, and methodology. Substantial revisions are required.

1. Line 19, CPVID-19 has become a pandemic. “is therefore now considered a” is an inaccurate sentence. Further, this is also a wrong sentence: is therefore considered as.
2. Line 20-21, clinical information is highly concerned, a unclear sentence.
3. Line 31, what is TT?
4. Line 27-32, in parentheses, somewhere the authors used P values, some where the authors used proportions. Please provided exact P values and main statistics.
5. Line 35, “worse physical condition” is overlapped with “multiple comorbidities”.
6. Line 36, “a high-risk for fatal complications”, sorry, I did not see any results on fatal complications in the above paragraph. Further, this is a wrong sentence, should be at high risk for.
7. For the main text, please provide the correct citation and reference format according to authors’ guideline of PeerJ. Can be a revised paper poor like this? The authors can not revise their paper in this way. If you did not care this paper, I and the reviewers have no any reasons to care this paper.
8. Line 46-47, please update these figures accordingly. The total number of infections has exceeded 10 million.
9. Line 49, “the initial COVID-19 cases..” is a wrong sentence. Even the authors upload a certification for language editing, I can not agree with the quality of language of this paper.
10. Line 51-54, a long and redundant sentence.
11. Introduction is inadequate. As pointed out by one of our reviewers, there have been a lot of similar studies. The authors need to review these literature and clarify why the current study is necessary. The authors just deleted the previous sentence describing very few related studies. Can the paper be treated like this? A waste of the precious time of our esteem reviewers.
12. Methodology. This is a case-control study but the authors did not specify how the control group was selected, for example, how age and gender was matched.

Issues of this paper are not only limited to the above I found, I strongly suggest the authors, please, revise it with your heart. An extensive peer review is necessary for this paper.

·

Basic reporting

In general, the quality of the paper and the English have been improved.
Please, see line 98, use "admission" instead of "admition"

Experimental design

As suggested, the study population has been better described.
In addition, I'd like to know how many HD patients with COVID-19 the authors have seen and how they selected the 16 patients included in this analysis.

Validity of the findings

Conclusions have been changed respect to the first version of the paper and now they are more pertinent to the data presented.

Additional comments

The quality of the study has been improved, such as its clarity.

Reviewer 3 ·

Basic reporting

Further Reviewer Comments to R1
Original Comment-2: Hemodialysis patients were a high-risk group of SARS-coV-2 infections. As shown in PubMed, more than 70 papers concerning COVID-19 in Hemodialysis patients have been published since the outbreak of COVID-19. However, the authors claimed several times that "Currently, the research about hemodialysis patients with COVID-19 is scarce. 143 This study is one of the first to report clinical characteristics and outcomes exhaustively". The authors did not search in Pubmed? Compared with these published papers, the current manuscript added little to this area.
Author Response-2: We appreciate the views of the reviewer. We read through the full text, corrected errors, and re-edited some sentences. For specific modifications, please see the revised manuscript.

Further Reviewer Comments to this reply: The authors should clearly clarify what’s they added to this area.

Comment-4:The diagnostic criteria was from WHO, but the treatment was using Chinese guideline 6th edition?
Response-4: We thank the reviewer for their detailed and professional comments. As mentioned by reviewer, for COVID-19 patients, especially hemodialysis COVID-19 patients, the medication treatment approach may be different from other countries or regions, or it is well outside the standard of care practiced in the United States and other nations. In particular, we introduced traditional Chinese medicine treatment for different patients' conditions (Table S2). Therefore, the treatment is to use the 6th edition of the Chinese Guidelines.
Further Reviewer Comments to this reply: The authors should check and make sure they used WHO diagnostic criteria while not the local criteria of China.

Experimental design

Further Reviewer Comments to R1
Original Comment-2: Hemodialysis patients were a high-risk group of SARS-coV-2 infections. As shown in PubMed, more than 70 papers concerning COVID-19 in Hemodialysis patients have been published since the outbreak of COVID-19. However, the authors claimed several times that "Currently, the research about hemodialysis patients with COVID-19 is scarce. 143 This study is one of the first to report clinical characteristics and outcomes exhaustively". The authors did not search in Pubmed? Compared with these published papers, the current manuscript added little to this area.
Author Response-2: We appreciate the views of the reviewer. We read through the full text, corrected errors, and re-edited some sentences. For specific modifications, please see the revised manuscript.

Further Reviewer Comments to this reply: The authors should clearly clarify what’s they added to this area.

Comment-4:The diagnostic criteria was from WHO, but the treatment was using Chinese guideline 6th edition?
Response-4: We thank the reviewer for their detailed and professional comments. As mentioned by reviewer, for COVID-19 patients, especially hemodialysis COVID-19 patients, the medication treatment approach may be different from other countries or regions, or it is well outside the standard of care practiced in the United States and other nations. In particular, we introduced traditional Chinese medicine treatment for different patients' conditions (Table S2). Therefore, the treatment is to use the 6th edition of the Chinese Guidelines.
Further Reviewer Comments to this reply: The authors should check and make sure they used WHO diagnostic criteria while not the local criteria of China.

Validity of the findings

Further Reviewer Comments to R1
Original Comment-2: Hemodialysis patients were a high-risk group of SARS-coV-2 infections. As shown in PubMed, more than 70 papers concerning COVID-19 in Hemodialysis patients have been published since the outbreak of COVID-19. However, the authors claimed several times that "Currently, the research about hemodialysis patients with COVID-19 is scarce. 143 This study is one of the first to report clinical characteristics and outcomes exhaustively". The authors did not search in Pubmed? Compared with these published papers, the current manuscript added little to this area.
Author Response-2: We appreciate the views of the reviewer. We read through the full text, corrected errors, and re-edited some sentences. For specific modifications, please see the revised manuscript.

Further Reviewer Comments to this reply: The authors should clearly clarify what’s they added to this area.

Comment-4:The diagnostic criteria was from WHO, but the treatment was using Chinese guideline 6th edition?
Response-4: We thank the reviewer for their detailed and professional comments. As mentioned by reviewer, for COVID-19 patients, especially hemodialysis COVID-19 patients, the medication treatment approach may be different from other countries or regions, or it is well outside the standard of care practiced in the United States and other nations. In particular, we introduced traditional Chinese medicine treatment for different patients' conditions (Table S2). Therefore, the treatment is to use the 6th edition of the Chinese Guidelines.
Further Reviewer Comments to this reply: The authors should check and make sure they used WHO diagnostic criteria while not the local criteria of China.

Additional comments

Further Reviewer Comments to R1
Original Comment-2: Hemodialysis patients were a high-risk group of SARS-coV-2 infections. As shown in PubMed, more than 70 papers concerning COVID-19 in Hemodialysis patients have been published since the outbreak of COVID-19. However, the authors claimed several times that "Currently, the research about hemodialysis patients with COVID-19 is scarce. 143 This study is one of the first to report clinical characteristics and outcomes exhaustively". The authors did not search in Pubmed? Compared with these published papers, the current manuscript added little to this area.
Author Response-2: We appreciate the views of the reviewer. We read through the full text, corrected errors, and re-edited some sentences. For specific modifications, please see the revised manuscript.

Further Reviewer Comments to this reply: The authors should clearly clarify what’s they added to this area.

Comment-4:The diagnostic criteria was from WHO, but the treatment was using Chinese guideline 6th edition?
Response-4: We thank the reviewer for their detailed and professional comments. As mentioned by reviewer, for COVID-19 patients, especially hemodialysis COVID-19 patients, the medication treatment approach may be different from other countries or regions, or it is well outside the standard of care practiced in the United States and other nations. In particular, we introduced traditional Chinese medicine treatment for different patients' conditions (Table S2). Therefore, the treatment is to use the 6th edition of the Chinese Guidelines.
Further Reviewer Comments to this reply: The authors should check and make sure they used WHO diagnostic criteria while not the local criteria of China.

---

## Round 0.3 · Major Revisions

I do not think the English language of this revision is acceptable, although the paper has been edited by EditSprings. I strongly suggest the authors revise the language of the paper again. The authors may consider using the language-editing services provided by other services providers such as the service offered by PeerJ.

·

Basic reporting

The paper has been improved, but some language problems persist. I advise a new revision by an English-native speaker.

Experimental design

Clear and well-explained.

Validity of the findings

The data are well-presented.

Additional comments

The paper has been improved after revision steps. However, these data are not so original, since larger studies of the same topic have already been published.
Nevertheless, the paper may be of interest to clinical nephrologists.

---

## Round 0.4 · accepted · Accept

This version seems much better. The paper can be accepted now.